# Optical imaging of localized chemical events using programmable diamond quantum nanosensors

Torsten Rendler[1,*], Jitka Neburkova[2,3,*], Ondrej Zemek[4], Jan Kotek[4], Andrea Zappe[1], Zhiqin Chu[1], Petr Cigler[2] & Jörg Wrachtrup[1]

Development of multifunctional nanoscale sensors working under physiological conditions enables monitoring of intracellular processes that are important for various biological and medical applications. By attaching paramagnetic gadolinium complexes to nanodiamonds (NDs) with nitrogen-vacancy (NV) centres through surface engineering, we developed a hybrid nanoscale sensor that can be adjusted to directly monitor physiological species through a proposed sensing scheme based on NV spin relaxometry. We adopt a single-step method to measure spin relaxation rates enabling time-dependent measurements on changes in pH or redox potential at a submicrometre-length scale in a microfluidic channel that mimics cellular environments. Our experimental data are reproduced by numerical simulations of the NV spin interaction with gadolinium complexes covering the NDs. Considering the versatile engineering options provided by polymer chemistry, the underlying mechanism can be expanded to detect a variety of physiologically relevant species and variables.

[1] 3. Physikalisches Institut, Universität Stuttgart, Pfaffenwaldring 57, 70569 Stuttgart, Germany. [2] Institute of Organic Chemistry and Biochemistry of the CAS, Flemingovo nam. 2, 166 10 Prague 6, Czech Republic. [3] First Faculty of Medicine, Charles University, Katerinska 32, 121 08 Prague 2, Czech Republic. [4] Faculty of Science, Department of Inorganic Chemistry, Charles University, Hlavova 2030, 128 43, Prague 2, Czech Republic. * These authors contributed equally to this work. Correspondence and requests for materials should be addressed to Z.Q.C. (email: z.chu@physik.uni-stuttgart.de) or to P.C. (email: cigler@uochb.cas.cz).

Physiological processes inside a living cell are accompanied by transient changes in variables including concentrations of ions[1], reactive oxygen species[2], enzymes[3], nucleic acids[4], pH[5] and redox potential[6]. Although a vast range of sensing principles for these variables based on selective molecular and nanoparticle probes have been developed[7–9], the field is still limited by the chemical and optical stabilities of probes, probe toxicity and perturbation of the biological environment, and, above all, probe sensitivity and spatiotemporal resolution. Therefore, reliable intracellular sensors designed for non-invasive quantitative monitoring of physiologically relevant species with near-atomic resolution are urgently needed to elucidate critical underlying mechanisms in cell biology and physiology, which in turn may lead to new possibilities for diagnostics and therapeutics at the subcellular level.

Nanodiamonds (NDs; nanometre-sized diamond particles) with nitrogen-vacancy (NV) defect centres exhibit excellent biocompatibility[10–12], long-term stability[13,14] and unique quantum sensing capability by optical means[15–17]. These NDs hold great promise for a range of biomedical applications, including serving as nanomedicine platforms for delivery of drugs[18], genes and proteins[19,20], use as fluorescent/photoacoustic imaging agents[13,21,22] and applications in multifunctional intracellular sensing[23–25]. The fluorescence of these atomic-scale NV centres in NDs depends on their electronic spin states, which show a long coherence time even under ambient conditions, enabling direct nanoscale sensing for magnetic/electric field[15,26], temperature[24,27,28] and mechanical force/pressure[29,30]. In fact, the facile optical readout of NV centres in NDs facilitates quantum sensing in living cells[23,24]. However, direct measurement of chemical reactions and processes through quantum detection of NDs remains challenging, especially under physiological conditions. The main challenges lie in developing selective detection principles enabling direct quantum sensing of chemical transformations via spin-dependent fluorescence of NV centres, the related chemical architectures on the ND surface that host the primary sensing system and robust quantum-sensing schemes applied to NV centres in NDs, especially when they are introduced to complicated environments such as the interior of living cells.

A critical step towards developing NDs with biomedical applications is customizing the diamond surface chemistry for required functionalization, while maintaining excellent colloidal stability under physiological conditions[31]. NDs engineered with our recently developed polymer-coating approach[32,33] exhibit long-term colloidal stability, reduced nonspecific binding and the capability for convenient chemical modification. In the current study, we connected macrocyclic complexes of $Gd^{3+}$ ions with a biocompatible copolymer shell on NDs via selectively cleavable linkers. In this sense, the attachment of $Gd^{3+}$ complexes to the polymer is strictly programmed, paving the way for their subsequent detachment in response to changes in a sole parameter. By quantifying the change in NV spin relaxation time due to the $Gd^{3+}$ complexes (spin noise), we show that this platform can be chemically programmed to sense fundamental physiological quantities. We designed and demonstrated time dependent pH and redox potential detection in a microfluidic device with sub-micrometre spatial resolution and minute temporal resolution. In particular, the excellent agreement between our experimental data and theoretical modelling suggests that this scheme can serve as a multifunctional platform for sensing of various chemical and biochemical transformations under physiological conditions with high selectivity (enabled by available libraries of selective cleavage reactions) and unprecedented sensitivity and resolution (yielded by the quantum detection approach).

## Results

**Design of ND-polymer-Gd multifunctional nanosensors.** To enable direct selective quantum detection of chemical processes by means of NV centres, we designed a general nanosensing platform that combines NV centres in NDs and surface polymer coating bearing spin labels. Specifically, complexes of $Gd^{3+}$ ions with electronic spin $S = 7/2$ were chemically attached via selectively cleavable linkers to poly[(2-hydroxypropyl)methacrylamide]-based (HPMA) co-polymer chains. Coating of NDs with an HPMA co-polymer shell improves the colloidal stability of the particles, reduces nonspecific interactions with proteins under physiological conditions, maintains the optical properties of NDs and enables further chemical modification[32,33]. The vicinity of $Gd^{3+}$ complexes (spin labels) acting as stochastically fluctuating magnetic fields can be sensed by NV relaxometry[34–37], providing us a novel route to monitor local

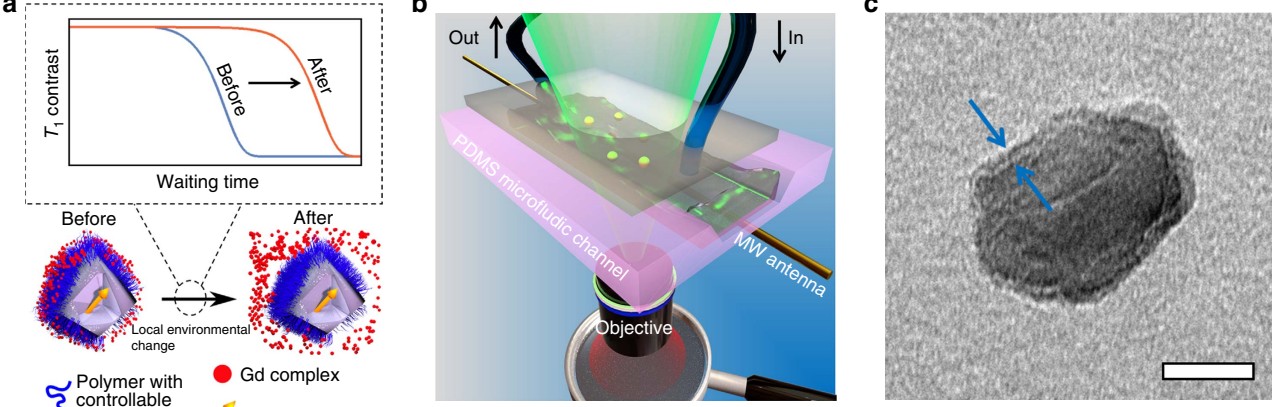

**Figure 1 | Basic principle of a ND-based multifunctional sensor.** (**a**) Cartoon showing the sensing mechanism of a ND-polymer-Gd hybrid nanosensor in response to a local environmental change. The $Gd^{3+}$ complexes (spin labels attached to the polymer shell on the ND surface) are released after activation of a chemical switch due to a local change, which can be monitored by the change in $T_1$ relaxation time of NV centres. (**b**) Cartoon showing the experimental setup: polydimethylsiloxane (PDMS) microfluidic channel (pink), pipe system enabling real-time measurement (blue), microwave antenna (gold), optical excitation for NV (green) and fluorescence detection (red). (**c**) Representative TEM image of a polymer-coated ND (ND-HPMA); arrows indicate the polymer shell. Scale bar, 20 nm.

changes. As illustrated in Fig. 1a, the $T_1$ relaxation time of NV centres in NDs can be quantitatively modulated by the surrounding $Gd^{3+}$ complexes: the more $Gd^{3+}$ complexes loading inside the surface polymer shell, the shorter the $T_1$ relaxation time. Detachment of $Gd^{3+}$ complexes from the ND particle strongly influences $T_1$ and can proceed only upon selective cleavage of the linker connecting the complex with the polymer. Importantly, we utilized complexes of $Gd^{3+}$ ions with macrocyclic ligands bearing one phosphonate/phosphinate and three acetate groups, which were originally designed as magnetic resonance imaging (MRI) contrast agents[38]. Thanks to their kinetic inertness and thermodynamic stability, these complexes do not release toxic $Gd^{3+}$ ions under physiological conditions and therefore exhibit excellent biocompatibility and negligible toxicity[39,40]. We developed a convenient synthetic pathway to their modification with cleavable linkers terminated with azido group for attachment to polymers via click chemistry (see Supplementary Methods). To demonstrate the potential applications of the developed nanosensor in cell biology and analytical chemistry, we performed experiments in a home-built microfluidic device made of polydimethylsiloxane (Fig. 1b). The NDs used are in average $\sim 33$ nm in diameter with a fairly narrow size distribution[41] (Supplementary Fig. 1) and coated with a HPMA shell a few nanometre thick, as indicated by transmission electron microscopy (TEM) image (Fig. 1c). These NDs contain few NV centres ($<10$ per particle on average, Supplementary Fig. 2).

**Chemical engineering and characterization of the ND surface.**
To test the utility of our ND-polymer-Gd hybrid nanosensors, we developed two kinds of chemical linkers to sense pH and redox potential, and corresponding nanosensors are denoted as ND@pH

and ND@redox particles, respectively (Fig. 2a). ND@pH particles contain an aliphatic hydrazone linker, for which the rate of hydrolytic cleavage is greatly accelerated at lower pHs in the physiologically relevant range[42] (pH 4–8; Fig. 2b). ND@redox particles contain a disulfide linker that can be cleaved into two thiol fragments in reducing environments (in this case, by the presence of glutathione, GSH) (Fig. 2c). As a control, we also synthesized a system with non-cleavable bonds (denoted ND-HPMA-Gd; Fig. 2a). The HPMA polymer is electroneutral but the macrocyclic $Gd^{3+}$ complexes are negatively charged, introducing an overall negative charge to the polymer shell. The electrostatic repulsion between complexes facilitates the departure of the cleaved complexes from the shell.

Based on the NV relaxometry-sensing scheme, the $T_1$ relaxation time of an NV centre is determined by the number of spins within the effective NV-sensing radius[35]. Therefore, the critical parameter is the actual concentration of $Gd^{3+}$ in the ND nanoenvironment as the relaxation time scales with the $Gd^{3+}$ concentration in the shell (see Supplementary Equations (2, 7, 12)). Either swelling or collapse of the polymer shell would affect this quantity and therefore influence the measured $T_1$ relaxation time, even if the $Gd^{3+}$ complexes are not released. As polymers can reversibly respond to pH and ionic strength by changes in their hydrodynamic diameters and also by nonspecific adsorption of ions resulting in changes in zeta potential, we studied the influence of these parameters on the behaviour of our nanosensors. First, we measured the size distribution of ND-HPMA-Gd particles (with non-cleavable bonds) in various pH buffers by dynamic light scattering. As shown in Fig. 2d, the hydrodynamic radii of ND-HPMA-Gd in the whole range of buffers were fairly uniform, indicating that the shell thickness of our nanosensor does not change in response to various buffer conditions. In two selected buffers (pH 2.0 and 7.4), we measured

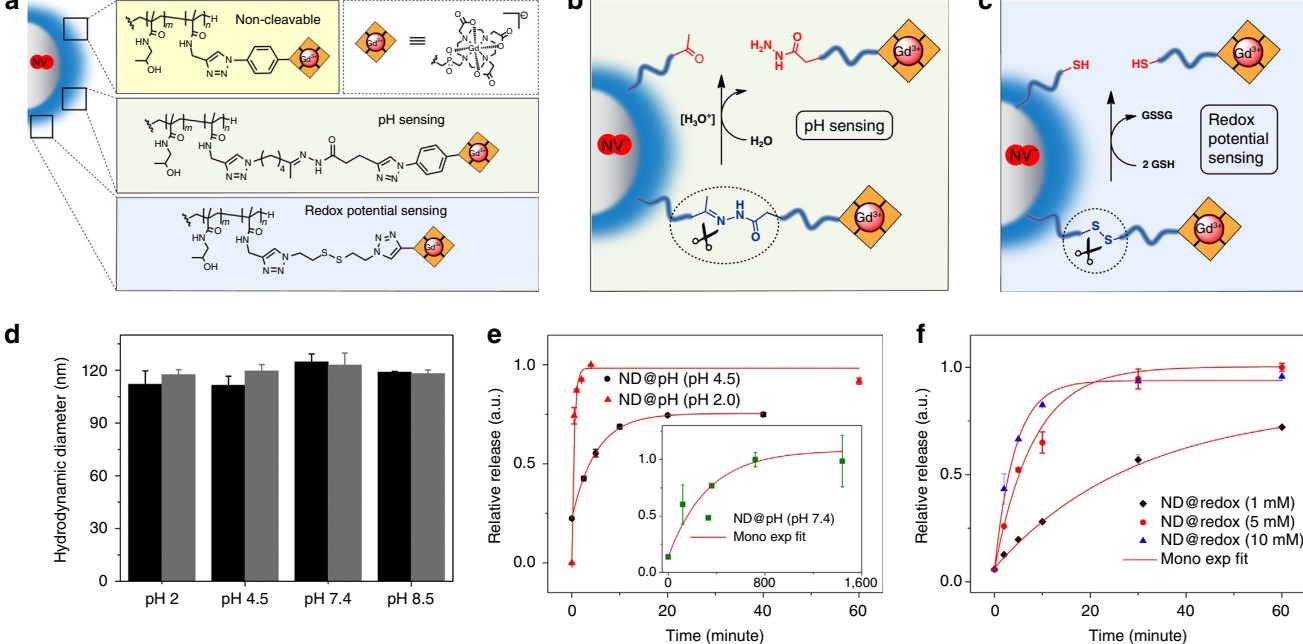

**Figure 2 | Design and characterization of ND-polymer-Gd hybrid nanoscale sensors.** (**a**) Chemical structure of the polymer interface with $Gd^{3+}$ complexes attached via a non-cleavable and two types of cleavable linkers. The specific release mechanisms for (**b**) pH-dependent hydrolytically cleavable and (**c**) reductively cleavable linkers are shown in detail. (**d**) Hydrodynamic diameters of poly(HPMA)-coated NDs (ND-HPMA, black) and poly(HPMA)-coated NDs modified with non-cleavable $Gd^{3+}$ complexes (ND-HPMA-Gd, grey) determined by dynamic light scattering in different buffers used for Gd-release measurements. (**e**) Release kinetics of $Gd^{3+}$ complexes in ND@pH particles in pH 2.0, 4.5 and 7.4 buffers analysed by ICP MS. The red line is the corresponding mono-exponential fitting. (**f**) Release kinetics of $Gd^{3+}$ complexes in ND@redox particles in the presence of 1, 5 and 10 mM GSH in pH 8.5 buffer analysed by ICP MS. The red line is the corresponding mono-exponential fitting. The error bars in **d–f** represent s.d. from at least three independent measurements.

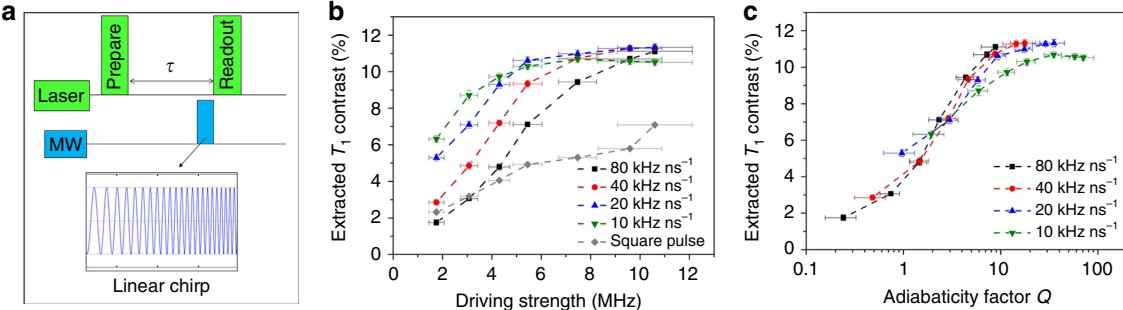

**Figure 3 | Robust relaxation measurement with linear chirp pulse.** (**a**) Schematic cartoon showing the used optical and microwave pulse sequence for $T_1$ relaxation measurement. (**b**) Comparison of experimentally extracted contrast of $T_1$ relaxation measurement with chirp pulse and square pulse for control sample ND-HPMA. The contrast is defined as the normalized initial difference between the sensing sequence with and without the inversion pulse. (**c**) Dependence of extracted $T_1$ contrast on the extracted minimum adiabaticity factor $Q$, where $Q$ is defined as the effective driving amplitude over its angular velocity in the rotating frame. The vertical error bars in **b,c** represent the s.d. from 20 independent measurements with $\tau \ll T_1$, whereas the horizontal error bars represent the s.e. (95% confidence intervals) from Lorentzian fits (for Fourier transformed Rabi oscillation).

hydrodynamic radii, zeta potentials and $T_1$ for ND-HPMA-Gd at increasing ionic strengths (up to ∼0.25 M, achieved by addition of NaCl and precisely quantified according to the conductivity of the solution). We observed that both pH and ionic strength exert only a marginal influence on hydrodynamic radii (Supplementary Fig. 3a). Although the zeta potentials increased with increasing ionic strength, most likely because of preferential adsorption of sodium ions (Supplementary Fig. 3b), this effect had no influence on $T_1$ (Supplementary Fig. 3c). This observation confirms the insensitivity of the polymer surface architecture to various environments, which is essential for the construction of stable and robust nanosensors for bio-applications.

To verify whether our nanosensors could release $Gd^{3+}$ complexes on demand, we incubated them in different environments, removed them from solution by centrifugation and analysed the amount of $Gd^{3+}$ complexes in the supernatant with inductively coupled plasma mass spectrometry (ICP MS) as a function of incubation time. This method allowed us to study the temporal evolution of $Gd^{3+}$ complexes release from our nanosensors. As shown in Fig. 2e, the ND@pH particles showed pH-dependent release, whereas the ND-HPMA-Gd (with non-cleavable bonds) particles (Supplementary Fig. 4) were stable under all conditions examined within the current chosen measurement time window (∼ 1 h). Furthermore, the slope of the release profile measured at pH 2.0 was two orders of magnitude higher than that for pH 7.4, indicating a much faster cleaving rate of $Gd^{3+}$ complexes from the polymer at low pH values. The very slow change (as shown in insert of Fig. 2e) at pH 7.4 indicates that the sensor can continuously operate for several hours before measurements at lower pH values.

Similarly, the ND@redox particles showed an obvious GSH concentration-dependent release: the higher the applied concentration of GSH, the steeper the release slope (Fig. 2f), whereas the ND-HPMA-Gd control sample (with non-cleavable bonds) were stable in the presence of GSH (Supplementary Fig. 5). All the measured release kinetics for ND@pH and ND@redox particles fit well with standard first-order reaction kinetics

$$C = C_0 e^{-kt} \qquad (1)$$

where $C_0$ is the initial concentration of the reactant and $k$ is the first order rate constant, indicating that the release rate is solely dependent on one specific reactant in solution.

### Robust NV spin relaxometry utilizing a linear chirp pulse.
Our experiments are based on probing the spin relaxation time $T_1$ of the NV centre. $T_1$ is measured by first initializing the NV spin

into $m_s = 0$ by an optical pulse. After a waiting time $\tau$, the spin state is readout by an optical excitation pulse generating fluorescence, which is proportional to the population probability of $m_s = 0$. Measuring this fluorescence as a function of the waiting time $\tau$ thus determines $T_1$. However, special care needs to be taken when performing relaxometric measurements on NV centres, as, for example, charge-state fluctuations can mask the $T_1$ decay in this approach[43]. To derive pristine $T_1$ curves, one needs to apply an additional microwave pulse in resonance with for example the $m_S = 0$ to $m_S = +1$ spin transition in a second measurement to invert the population of spin state sublevels and subtract the result of both (see Supplementary Discussion). As the spin state has to be manipulated, knowledge about the spin resonance frequency is required. In addition, the excitation power and pulse duration of the used microwave pulse has to be set correctly for a precise state adjustment. Later parameters for their part differ when the orientation of the NV axis to the local microwave field is changed. Therefore, it is highly desirable to use pulse schemas that intrinsically compensate for such variations. In the current study, we optimized the $T_1$ readout by introducing an adiabatic passage (see Supplementary Discussion) in form of a linear chirp pulse (Fig. 3a), which is robust against detuning and microwave driving power[44]. To compare the performance of different schemes used in $T_1$ measurement, we extracted the spin contrast from experimental measurement as a function of driving strength (estimated by Rabi oscillations driven on the optically detected magnetic resonance (ODMR) transitions, see Supplementary Fig. 10a) for the control sample ND-HPMA (without $Gd^{3+}$ complexes) (Fig. 3b). The obtained spin contrast decreased as the driving strength reduced for both measurement schemes, but the chirp pulse always resulted in a higher contrast than that obtained with square pulse. We also plotted the extracted spin contrast as a function of expectable minimal adiabaticity factor $Q$ (a measure of the adiabaticity of the used pulse scheme[45]) when using linear chirp pulse at different chirp rate (Fig. 3c). For the data shown in this work using linear chirp pulse, we experimentally obtained a contrast ranging from 4 to 10%, corresponding to a factor $Q$ in the range of around 1.5 up to 10. Indeed, in terms of sensitivity, the chirp pulse resulted in a twofold sensitivity enhancement with different chirp rate when the microwave power is large enough (Supplementary Fig. 6).

### Microfluidic measurement of nanosensors.
Because of the excellent colloidal stability of our nanosensor under physiological conditions, we performed our $T_1$ measurements on diffusing NDs (Fig. 4a). We first investigated two kinds of control sample,

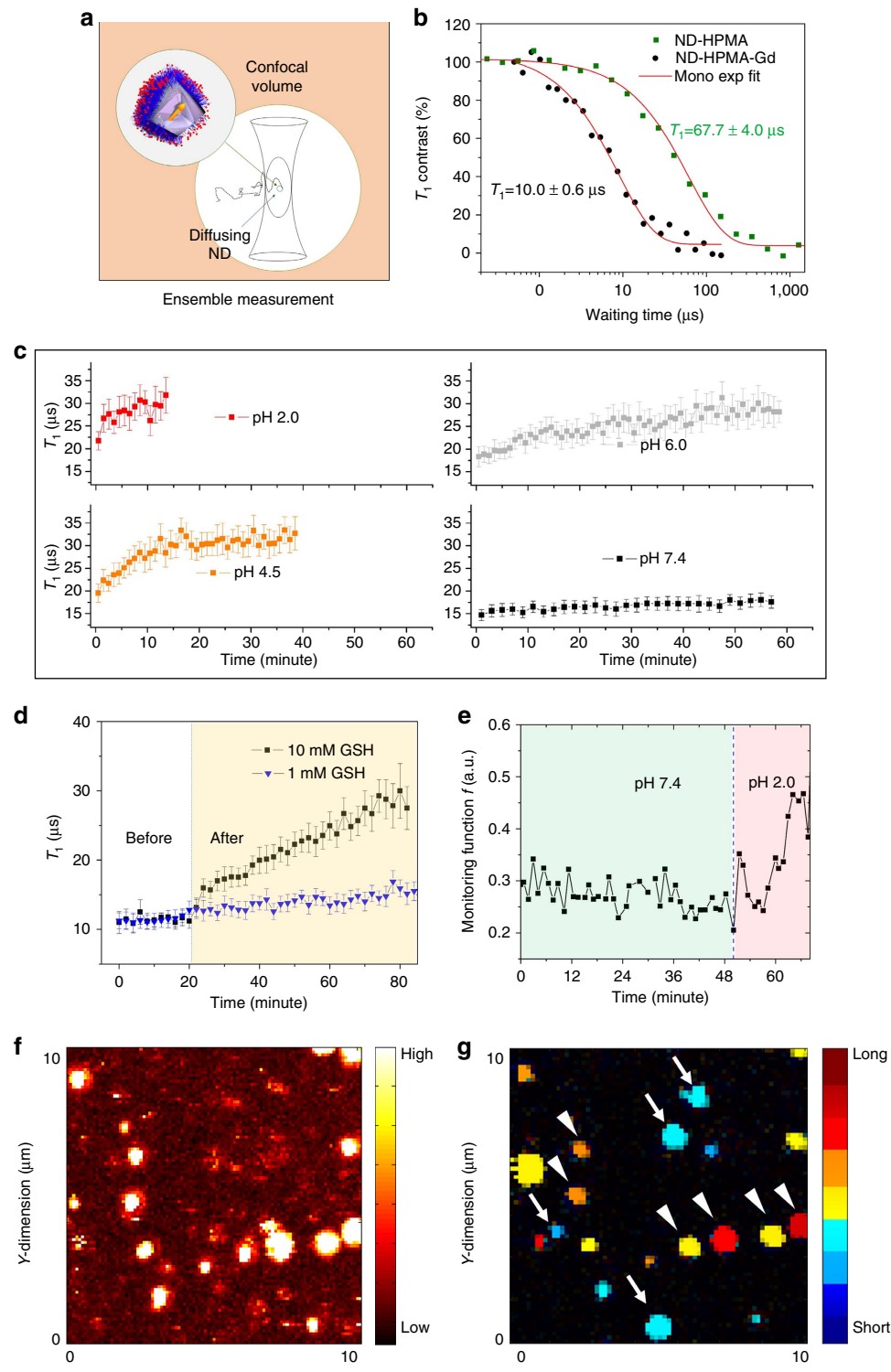

**Figure 4 | *In situ* measurements in a microfluidic channel.** (**a**) Schematic cartoon showing the ensemble measurements on the averaged $T_1$ of ND particles diffusing through the confocal volume. (**b**) Typical $T_1$ ensemble measurement of ND-HPMA and ND-HPMA-Gd (with non-cleavable bond) particles in pH 7.4 buffer. (**c**) Time-dependent ensemble measurement for $T_1$ of the ND@pH particles when incubated with pH 2.0, 4.5, 6.0 and 7.4 buffers. (**d**) Time-dependent ensemble measurement for $T_1$ of the ND@redox particles in buffer solution before and after addition of 1 mM (blue) and 10 mM (black) GSH. (**e**) Time-dependent fixed-τ measurement for chosen ND@pH particles when incubated at pH 7.4 followed by a change to pH 2.0 buffer (details are given in Methods). (**f**) Confocal image for the chosen view of ND@pH particles on cover glass in pH 2.0 buffer; the bar indicates the measured fluorescence intensity. (**g**) Reconstructed $T_1$ contrast image of the same view as in **f** after rinsing with pH 7.4 buffer and adding again freshly prepared ND@pH particles (loaded with $Gd^{3+}$ complexes) in pH 7.4 buffer. White triangles point to old ND@pH particles ($Gd^{3+}$ complexes released), while white arrows point to those newly emerging ones (loaded with $Gd^{3+}$ complexes), the colour bar indicates the extracted $T_1$ value ranging from short (blue) to long (red). The error bars in **c** and **d** represent the s.e. (95% confidence intervals) from mono exponential decay fits.

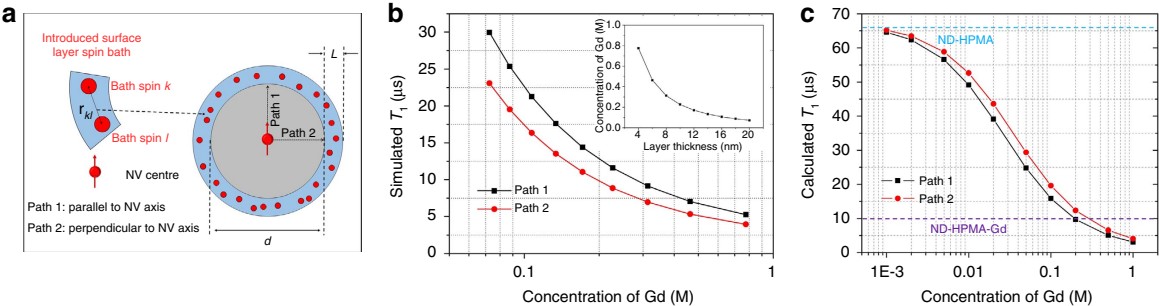

**Figure 5 | Simulation for the influence of Gd³⁺ complexes on relaxometry.** (**a**) The ND is modeled as a sphere with diameter $d$ coated with polymer with thickness $L$. The $Gd^{3+}$ complexes embedded in the polymer layer acts as a randomly fluctuating spin bath. The NV centre is positioned along two orientations (Path 1 and 2). (**b**) The influence of varying HPMA layer thickness (varying concentration of $Gd^{3+}$ complexes) on simulated $T_1$ value when the number of $Gd^{3+}$ complexes is fixed. The inset shows the concentration of $Gd^{3+}$ complexes versus the layer thickness of the shell (here, $d = 33$ nm and the number of $Gd^{3+}$ complexes was fixed at ~8,000). (**c**) The influence of varying concentrations of $Gd^{3+}$ complexes on calculated $T_1$ value when the thickness of the polymer layer is fixed ($d = 33$ nm and $L = 10$ nm). The purple dashed line (ND-HPMA-Gd) and blue dashed line (ND-HPMA) denote the lower and upper limit for the measured $T_1$ value in current study, respectively.

namely ND-HPMA (without $Gd^{3+}$ complexes) and ND-HPMA-Gd (non-cleavable). As shown in Fig. 4b, the ND-HPMA particles had an average measured $T_1$ value of $67.7 \pm 4.0$ μs, whereas the ND-HPMA-Gd particle showed a significantly lower value of $10.0 \pm 0.6$ μs in pH 7.4 buffer. In fact, we also checked the stability of these control samples in all the buffer conditions used in current study (see Supplementary Figs 4 and 5).

We then used the ND@pH particles (pH cleavable) to perform time-dependent $T_1$ measurement in various conditions including pH 2.0, 4.5, 6.0 and 7.4 (Fig. 4c). At physiological pH environment (pH 7.4), we observed only a slight change for the measured $T_1$ over a period of 1 h, whereas the $T_1$ changed dramatically at acidic condition. The measured $T_1$ curve as a function of the incubation period showed the following behaviour (Fig. 4c): the lower the pH value the steeper the changes in $T_1$, being consistent with our previous ICP MS measurements (Fig. 2e). Taking a moderate pH (pH 4.5) as an example, the release kinetics of $Gd^{3+}$ complexes (after converting $T_1$ to a $Gd^{3+}$ concentrations using Supplementary Equations, also see Fig. 5) from ND@pH particles ($\tau_{rel} = 315.7 \pm 42.8$ s) agree excellently with ICP MS measurements ($\tau_{rel} = 306.4 \pm 9.9$ s) (Supplementary Fig. 7). Using set of Britton–Robinson buffers (with equal composition and ionic strength), we also investigated the sensitivity of our sensor in the physiologically relevant pH range (3.7–6.9) for a short measurement time frame (12 min). We estimated the $T_1$ change rate for each measured pH and found a monotonous dependence on pH (Supplementary Fig. 8) allowing for accuracy of a pH difference at least ~0.7.

Furthermore, we also investigated the change in $T_1$ of ND@redox nanosensor (cleavable at reduction conditions) in the presence of GSH, an important antioxidant found in most of the animal cells[46]. As shown in Fig. 4d, we observed a mild increase in the $T_1$ change rate after adding 1 mM GSH and a significant change for 10 mM GSH, while the measured $T_1$ value was constant before GSH addition.

We can reduce the used measurement schema into fast fixed-$\tau$ measurement by directly counting the ratio (monitoring function $f$) of fluorescence signal at two fixed time points[43]. A large monitoring function $f$ value indicates a longer relaxation time. As shown in Fig. 4e, the monitoring function $f$ kept almost the same in pH 7.4 buffer, but starts to increase once the pH 2.0 buffer is introduced, indicating the increase of $T_1$ time due to $Gd^{3+}$ complexes release (being consistent with data in Figs 2e and 4c). With the fast measurement scheme, we can perform $T_1$ contrast imaging on several different NDs in a confocal scanning

approach. To demonstrate this, we first incubated ND@pH particles in pH 2.0 buffer as shown in Fig. 4f, followed by rinsing with pH 7.4 buffer and adding again freshly prepared ND@pH particles in pH 7.4 buffer. Next, we performed fixed-$\tau$ measurement for each pixel within the chosen field of view and reconstructed a $T_1$ contrast image shown in Fig. 4g. ND@pH particles that were not subject to low pH (white arrows in Fig. 4g) showed short $T_1$ times, whereas particles that were subject to pH 2.0 (white triangles in Fig. 4g) showed long $T_1$ times, as most of their loaded $Gd^{3+}$ complexes had been released during pH 2.0 treatment. This is consistent with the statistical view of $T_1$ observations for the sample ND@pH, indicating significant different $T_1$ in two different pH buffers (pH 2.0 and pH 7.4; Supplementary Fig. 9).

**Modelling the NV relaxometry modulated by $Gd^{3+}$ complexes.** To fully understand the observed effects, we started with the model description[47] and revised it to describe the function of our nanosensors (see Supplementary Methods). As shown in Fig. 5a, we modelled a single ND particle as a sphere and considered the introduced $Gd^{3+}$ complexes as randomly fluctuating spin bath inside the polymer shell at the beginning. As the actual position of our NV centre in the crystal is not known, we considered various positions of the NV centre between the centre and the edge of the sphere. By this we derive an average value of the simulated $T_1$. From measurements using ICP atomic emission spectroscopy (AES), we obtained the average content of $Gd^{3+}$ complexes in ND particles ~3.2%. Therefore, we can estimate the approximate number of $Gd^{3+}$ complexes is around 8,000 per particle if we assume an average diameter of NDs as 33 nm, equivalent to the average value obtained from our TEM measurements (Supplementary Fig. 1). To estimate the shell thickness of the HPMA for the given sample in the used buffer solutions, we fixed the number of $Gd^{3+}$ complexes inside the polymer shell and varied its thickness to investigate the corresponding $T_1$ change. In this way, we derive the $T_1$ value as a function of shell thickness as shown in Fig. 5b. The measured $T_1$ of ND-HPMA-Gd is ~10 μs (Fig. 4b) pointing to ~0.2 M $Gd^{3+}$ concentration in our simulation (Fig. 5b) with a ~10 nm shell (insert in Fig. 5b). This shell thickness is similar with our TEM observation in Fig. 1c and Supplementary Fig. 1, and we used it in all further analysis steps. One could also account for paramagnetic centres lying on the surface of NDs[47], but as we know the relaxation rate for the situation when the $Gd^{3+}$ is absent, we can use the already gained $T_1$ time from the control sample ND-HPMA as a basis

offset. The measured $T_1$ value was 67.7 µs (Fig. 4b) for the control sample ND-HPMA in buffer solutions. We interpreted this as the intrinsic relaxation rate

$$R^{int} = \frac{1}{T_1^{int}} \qquad (2)$$

where $T_1^{int} = 67.7$ µs of the nanosensors in solution and calculated the expected relaxation $T_1$ time (Fig. 5c) using

$$T_1^{Calculated} = \left(R^{int} + R^{Gd}\right)^{-1} \qquad (3)$$

where $R^{int}$ is the intrinsic relaxation rate and $R^{Gd}$ is the simulated relaxation rate induced by $Gd^{3+}$ complexes. From this, we can deduct that we are able to detect $Gd^{3+}$ complexes with concentration ranging from 0.2 M ($\sim 8,000$ $Gd^{3+}$ complexes per particle) down to 0.001 M ($\sim 40$ $Gd^{3+}$ complexes per particle), corresponding to the measured highest (purple dashed line in Fig. 5c) and lowest (blue dashed line in Fig. 5c) Gd content, respectively.

## Discussion

Our hybrid nanosensor achieves signal transduction, recording and amplification simultaneously. Subtle changes in physiological systems (weak signals) can be recorded by counting the variance of $Gd^{3+}$ complexes (strong interaction with NV centres) inside the polymer shell due to a programmed chemical reaction. The well-fitted first-order reaction equation for measured release kinetics (Fig. 2e,f and Supplementary Fig. 7) indicates that our nanosensor responds to changes in a single, pre-defined chemical parameter. Importantly, the thickness of the polymer shell was insensitive to pH and ionic strength (Fig. 2d and Supplementary Fig. 3a). We observed neither swelling nor collapse of the polymer shell in any of the conditions used. Consistently, we observed no influence of these factors on $T_1$ (Supplementary Fig. 3c), which is critical for reliable and robust function of the nanosensors in biological environments.

The excellent agreement between the experimental results (Fig. 4b) and theoretical modelling (Fig. 5c) indicates the underlying mechanism: the change in $T_1$ relaxation time is caused by release of $Gd^{3+}$ complexes from the polymer shell. Precise agreement of ICP MS with $T_1$ kinetic measurements (Supplementary Fig. 7) suggests the high accuracy and sensitivity of the current detection method. In principle, we can monitor gradual release down to several tens of molecules of $Gd^{3+}$ complexes (Fig. 5c) at a single-particle level (Fig. 4e–g), which allows us to monitor a localized chemical process occurring on an extremely small scale ($\sim 10^{-22}$–$10^{-20}$ mol). Although the achieved accuracy ($\sim 0.7$ pH unit) is lower compared with the current, most sensitive measurement techniques[48] ($\sim 0.1$ pH unit for intracellular measurement), our system operates in quite a broad pH range. In contrast, for some optical pH sensors, which exhibit a sigmoidal response towards changes in pH, their narrow dynamic range represent often a limitation[49]. Considering practical measurements in cells, the accuracy of our sensor is sufficient, as pH differences between extracellular space, cytosol and some organelles are much higher than 0.7. For example, the cytosol pH is $\sim 7.4$, whereas endo/lysosomal compartments show pH $\sim 4.5$ (ref. 50). Similarly, the intracellular GSH concentrations usually range from 0.5 to 10 mM, whereas extracellular values are almost three orders of magnitude lower[51]. These differences are in a range well measurable by our nanosensor.

Many of the currently used nanosensors are based on mechanisms, which are either irreversible (based on formation or cleavage of covalent bonds) or practically irreversible, because the formed non-covalent sensing assembly is extremely stable (for instance, nucleic acid hybridization, antibody and aptamer affinity probes, fluorescence resonance energy transfer sensors utilizing cleavage reactions[52]). Irreversibility is typical also for current approaches to detect GSH[53,54]. The chemical nature of our sensing mechanism also renders our scheme irreversible, which limits the possible durations of measurements, especially for higher cleavage rates. To enlarge the measurement window to basically unlimited time, we are currently developing a reversibly responding polymer coating on NDs, which operates without a need of irreversible cleavage events.

For a typical $T_1$ measurement with an additional control sequence using a square pulse, one needs to find the resonance frequency and the length of used pulse to effectively invert the spin population. This is especially important for NDs, because their NV centres are typically arbitrary oriented and with a strain-induced variation in resonance frequency[23]. In comparison, the used linear chirp pulse scheme simplifies $T_1$ measurements into a single step: direct $T_1$ measurement by applying chirp pulse acting as 'inversion pulse' without any preliminary measurements for identifying the resonance frequency and pulse length. In addition, the chirp pulse scheme results in enhanced sensitivity compared to that with a square pulse (Fig. 3 and Supplementary Fig. 6). This is also consistent with our simulation of NV spin state evolution excited by different pulse scheme: the square pulse is sensitive to the changes in microwave excitation while the linear chirp pulse is much more robust, especially in the presence of inhomogeneous ODMR line broadening (see Supplementary discussion and Supplementary Fig. 10). Thus, the chirp pulse scheme enables robust $T_1$ measurement on different NDs simultaneously (ensemble measurement).

In conclusion, our hybrid nanosensor, owing to its versatility, can serve as a general platform with potential applications ranging from catalytic chemistry to cell biology and physiology, especially for label-free three-dimensional imaging of physiological variables by optical means. Development of molecular-sized NDs with NV centres[55,56] can further increase the sensitivity of the current method due to improved spin sensitivity of NV centres.

## Methods

**Experimental setup.** In the current study, we adapted a confocal microscopy apparatus. The laser (CNI, CW DPSS Laser 532 nm) was directed through acousto-optic modulator (AOM) and focused onto the focal plane of a $\times 60$ water-immersion objective (Olympus) for the ensemble measurements and a $\times 60$ oil objective (Olympus) when measuring individual NDs. The fluorescence of NV centre was filtered (long pass, cutoff at 647 nm) and collected by two avalanche photo diode (Perkin-Elmer) in Hanbury-Brown and Twist configuration. Resonant microwave manipulation of the NV centre was achieved using a spanned copper wire inside a home-built microfluidic channel made of transparent poly-dimethylsiloxane (Sylgard 184 silicone elastomer kit, Dow Corning) in the vicinity of the optical focus. Two small plastic tubes are used to exchange the solution in the microfluidic channel.

**Preparation of ND-polymer-Gd nanosensors.** Detailed descriptions for preparation of fluorescent NDs with NV centres[57], their coating with HPMA polymer and synthesis of $Gd^{3+}$ complexes can be found in the Supplementary Information. Briefly, alkyne-modified HPMA-coated NDs were decorated with azide-modified $Gd^{3+}$ complexes using Cu(I)-catalysed azide-alkyne cycloaddition (CuACC). HPMA-coated NDs (10 mg in a final reaction volume of 12.8 ml of 50 mM HEPES buffer, pH 7.4) were mixed with either non-cleavable $Gd^{3+}$ complexes or $Gd^{3+}$ complexes with hydrazone or disulfide linker (in final concentrations of 0.96, 1.92 and 2.4 mM, respectively), pre-mixed 0.32 mM $CuSO_4$ and 0.64 mM tris(3-hydroxypropyltriazolylmethyl)amine ligand, and a freshly prepared solution of sodium ascorbate (5 mM). The reaction mixture was well sealed, left for 1 h with no stirring and washed by centrifugation with water ($Gd^{3+}$ conjugates with non-cleavable linker, ND-HPMA-Gd and disulfide linker, ND@redox) or methanol ($Gd^{3+}$ conjugate with hydrazone linker, ND@pH). The resulting nanosensors were stored in water (ND-HPMA-Gd and ND@redox) or in dry methanol (ND@pH) at 4 °C.

**Characterization of ND-polymer-Gd nanosensors.** The morphology and size of the particles were characterized with TEM (JEOL JEM-1011)[58]. The stability and

surface charge of HPMA-coated NDs with $Gd^{3+}$ complexes were tested by dispersing them in buffer solutions (50 mM citric acid buffer pH 2.0, 50 mM acetate buffer pH 4.5, 50 mM HEPES buffer pH 7.4, 50 mM TRIS buffer pH 8.5 and 1.5 M PBS buffer pH 7.4) for further experiments. Dynamic light scattering and zeta potential were recorded with a Zetasizer Nano ZS system (Malvern Instruments) at 37 °C at a concentration of 0.1 mg ml$^{-1}$.

To quantitatively measure the amount of $Gd^{3+}$ complexes released from the nanosensors, the particles were mixed with buffer and incubated for a certain time. Then, cleavage conditions were stopped, the particles were centrifuged and the released $Gd^{3+}$ complexes in supernatant were measured with an ICP MS 7700 (Agilent Technologies) instrument in duplicates. The non-cleavable ND-HPMA-$Gd^{3+}$ conjugate was used as a control and processed under the same conditions. The relative release at a given time was calculated as a ratio of the amount released to the maximum release amount. A detailed description of these release experiments can be found in the Supplementary Information. The total amount of $Gd^{3+}$ conjugated to HPMA-coated NDs was measured as ∼3.2% (weight percentage to NDs) using ICP AES (Spectro Arcos SOP).

**Relaxation measurement with linear chirp pulse.** For full $T_1$ relaxometry measurement, laser light modulation was achieved by passing continuous wave laser through an acousto-optical modulator for polarization and readout of NV centres. We first applied a laser pulse for polarizing NV centres into $m_s = 0$ (initialization) and then wait for the time $\tau$, followed by another laser pulse to detect NV fluorescence revealing the spin state (readout). Afterwards, we applied a similar sequence that differs from the first one, by adding a microwave pulse before the readout. The microwave pulse (linear chirp) is generated by mixing the output of one microwave source (SMIQ 03B, Rhode & Schwarz) with an arbitrary waveform generator (AWG2041, Tectronixs) and amplified by a microwave amplifier (ZHL-16W-43+, mini circuits). The linear chirp microwave pulse starts from 2.845 GHz and is swept over 100 MHz (covering most of the detuning range in NDs) at certain speed. The chirp speed was kept as 10–100 kHz ns$^{-1}$. The obtained difference in fluorescence $\Delta F(\tau)$ is proportional to the residual spin polarization after time $\tau$ of only those NV centres excited by microwave pulse. We thus further normalized the obtained $\Delta F(\tau)$ (named $T_1$ contrast)[43], fitted it with a single exponential function:

$$\Delta F(\tau) = A e^{-\tau/T_1} \qquad (4)$$

to get an average decay constant $T_1$ value. For comparison of a chirp pulse with a square pulse for effective spin state inversion during $T_1$ measurement, 10 μl poly(HPMA)-coated NDs (4 mg ml$^{-1}$ in water) were dropped in the vicinity of the copper wire on top of cover glass and air dried. The analysed detection volume contains more than several hundreds of NV centres, estimated on the detected photon flux in this experiment in comparison with that detected from a single NV centre measured with the same setup. The focal point of the laser was tuned to any position of the dense packed NDs nearby the copper wire for ensemble measurements (all the NDs inside the focus volume). At different microwave power, we performed $T_1$ measurement through both square pulse and chirp pulse with different sweeping speed. The adiabaticity factor $Q$ is defined as the effective Larmor precession around the effective magnetic field in the rotating frame over the angular change of the field[45]. The driving strength is defined as the effective Rabi frequency of an ensemble of NV centres driven by an external microwave. To quantify the performance of individual scheme in an experiment, the sensitivity enhancement factor is calculated as ratio of power noise equivalents $\delta T_1$ of the different pulse scheme:

$$EF = \frac{\delta T_1^{chirp}}{\delta T_1^{square}} = \frac{c_{chirp}}{c_{square}} \cdot \sqrt{\frac{t_{square}}{t_{chirp}}} \qquad (5)$$

where $c$ is the contrast and $t$ is the cycle time of the measurement.

**NV relaxometry measurements in a microfluidic channel.** For all time-dependent $T_1$ measurements (Fig. 4 and Supplementary Fig. 9), the freshly prepared nanosensor particles were dispersed in the respective buffer at a concentration of approximately 100 μg ml$^{-1}$ and were injected into a microfluidic channel through the conjugated tube. In case of the ensemble measurement we used a PDMS chamber that can be opened and covered from top. The focal point of the laser was placed to any position inside the channel for ensemble measurements (all the free diffusing particles). For fixed $\tau$ measurement, we only collected the fluorescence signal ($F$) at two fixed time points ($\tau_1 = 0.001$ μs and $\tau_2 = 20$ μs) on the obtained full $T_1$ curve of chosen ND spot and compare the change of $T_1$ by monitoring function:

$$f = F(\tau_2)/F(\tau_1) \qquad (6)$$

**Data availability.** Data supporting the findings of this study are available within the article and its Supplementary Information files and from the corresponding authors upon reasonable request.

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

## Acknowledgements

The authors thank Philipp Neumann, Thomas Häberle and Florestan Ziem for fruitful discussions, Matthias Widmann for his assistance with drawing illustration cartoons in Fig. 1, Tomas Matousek for ICP MS measurements, and David Chvatil and Jan Stursa (Nuclear Physics Institute AS CR, Czech Republic) for irradiation of nanodiamonds, which was supported by grant project RVO61389005. P.C. and J.N. acknowledges the support from the Grant Agency of the Czech Republic (Project Number 16-16336S). J.K. and O.Z. acknowledge the support from the Grant Agency of the Czech Republic (Project Number 16-03156S). J.W. acknowledges funding by the DFG via SPP 1601 and FOR 1493, the Volkswagenstiftung and the EU via the IP DIADEMS.

## Author contributions

Z.Q.C., P.C. and J.W. conceived the idea. T.R. designed and carried out all the optical measurements. J.N., O.Z. and J.K. performed all materials synthesis and characterization. A.Z. prepared all the buffer solution involved in optical measurements and participated in the optimization for optical measurements. T.R. carried out the theoretical calculations. T.R., Z.Q.C. and P.C. analysed the data. J.W. supervised the project. Z.Q.C., P.C. and J.W. wrote the manuscript with the comments from all co-authors. All authors discussed the results and commented on the manuscript.

## Additional information

**Competing financial interests:** The authors declare no competing financial interests.

