## [Peer Review File · Nature Communications]

Reviewer #1 (Remarks to the Author):

Real-time optical imaging of localized chemical events using programmable diamond quantum nanosensors

Torsten Rendler, Jitka Neburkova, Ondrej Zemek, Jan Kotek, Andrea Zappe, Zhiqin Chu, Petr Cigler, Jörg Wrachtrup

Herewith I am reviewing the above mentioned manuscript (manuscript number NCOMMS-16-17056) which is under consideration to be published in Nature Communications. The article is about measurements of pH and redox potentials in the nanoscale based on NV defects in diamonds. It is easy to imagine fields where this could have a tremendous impact. Areas where this could be interesting are for instance cell biology or chemistry or material science in the nanoscale. To the best of my knowledge this has not been done so far and is definitely worth pursuing. The approach the authors followed was to connect macrocyclic complexes of Gd³⁺ ions with a biocompatible copolymer shell to NDs via selectively cleavable linkers. These linkers are then cleaved in response to for instance a pH change releasing the Gd from the diamond particle and increasing the coherence times. Depending on the application, these linkers can be different and thus be sensitive for either pH or redox potential.

The article is well written, clear and the overall quality is very good. Control experiments (as for instance ruling out effects on T₁ from swelling/shrinking of the polymer shell) were conducted carefully when needed.

The only concern that I have is that the changes in pH that were considered are quite drastic. So I would like the authors to comment on the resolution in pH/redox potential they can actually resolve. How much difference in pH/redox potential can you detect? However, since this application is so new and this is the first proof of principle data I think even if the sensors might not yet be fit to detect biologically relevant changes this is still a milestone in that direction. Overall this article is very good and I recommend to publish it (almost) as it is.

Reviewer #2 (Remarks to the Author):

In this paper, the authors demonstrate that slow changes in certain variables in the chemical environment can be transduced to changes in magnetic noise, which can subsequently be detected optically by using nitrogen-vacancy (NV) color centers in nanodiamonds. Specifically, nanodiamonds are functionalized with a co-polymer that binds to complexes carrying Gd³⁺ via a functional group, which in one case can be cleaved by changes in pH, and in another case - by changes in concentration of antioxidant GSH. The accelerated desorption of Gd³⁺ (the source of magnetic noise) from the functionalized nanodiamond surface is detected as a decrease in the relaxation rate of the NV magnetic sublevel populations (relaxometry).

This method of magnetic noise detection from Gd³⁺ (and other targets) via NV centers in diamond (including nanodiamonds) is well established (as in references 34,35,44 cited by the authors and [1,2] below which should also be cited). This type of co-polymer functionalization of nanodiamonds has also been shown before by multiple authors (including some of the current authors). The novelty in this work lies in combining the NV/Gd³⁺ relaxometry technique with nanodiamond functionalization unstable under certain chemical environments and thereby demonstrating the detection of the said chemical environments.

My main concern is the applicability of this sensing modality to detection and imaging of relevant chemical signaling in biological and chemical systems, especially with regards to the method's irreversibility and slow response (see specific comments on this in Major Comments below). I would like to see these comments/concerns addressed before recommending the paper for publication; perhaps they can be addressed with an appropriate discussion of the limitations of this

method. On another note, an additional control experiment is recommended.

Major Comments:

The system is out-of-equilibrium and the sensor operation is non-reversible. Namely, in the case of ND@pH, the polymer to Gd⁺ complex link is unstable on the time scale of hours even in neutral water, based on the data in the paper (Figures 2 and 4). Reduced pH simply accelerates the Gd⁺ desorption by a factor of few, depending on the pH. The same applies to increased GSH concentration. In other words, the authors don't actually measure pH or redox potential, but polymer cleavage rates, which are affected by these variables. This raises several concerns:

- 1) Only unidirectional changes in these variables can be detected (decrease in pH or increase in GSH).
- 2) There is not a one-to-one mapping between these chemical variables and the detected T1 signal; the signal depends on absolute time since the introduction of the sensor into the system and on the history of changes in these chemical variables.
- 3) The response of the signal to a fast change in pH or GSH is on a time scale of tens of minutes, which is too slow for many biological processes on the microscale and nanoscale.
- 4) As equilibrium is approached (the nanodiamonds loose all Gd⁺) on the hour-long time scale, the sensor stops working, which limits the length of time over which a system of interest can be monitored.
- 5) Since the authors present this as a sensing method, plots summarizing its sensitivity is in order, such as plots of time constant of T1 increase versus pH and GSH concentration, and perhaps plots of the slopes of the previous plots versus pH and GSH. How does this performance compare to other sensing modalities of pH and GSH in biological systems? What are the typical magnitudes of the changes in pH and GSH in biological systems, at the relevant pH and GSH values, and can this technique realistically be used to measure these changes (in terms of sensitivity - besides the time scale and irreversibility problems mentioned above)?

In view of comments 1-3 above, it is not very fair to call this technique time-resolved detection of chemical signals. Perhaps the authors can discuss situations where it would be fair and where these issues are mitigated. Alternatively, what are ways to mitigate them in future experiments?

The imaging part of the experiment, T1 was imaged over a surface covered with two types of nanodiamonds that underwent different acid treatment and hence had different Gd⁺ surface concentrations. Different T1 times were optically resolved on different nanodiamonds. A more thorough discussion is needed here. Namely, I have two concerns:

- 1) On the T1 image in Fig.4, only those nanodiamonds were highlighted where T1 times were long and the nanodiamonds were incubated in pH=2, or where T1 times were short and the nanodiamonds were incubated in pH=7. Clearly however, several nanodiamonds in that image do not show this correlation. To make clear statement about these results, a summary plot is in order, such as overlapped histograms of T1 times for the two types of nanodiamonds. What explains the apparently large variance?
- 2) Since the imaging was a static experiment performed after separate chemical treatments of different parts of the sample, it is unfair to make the claim of "real-time imaging of localized chemical events" in the paper and especially the title.

An important control experiment is missing: T1 measurements of non-cleavable nanoparticles ND-HPMA and ND-NPMA-Gd⁺ under the different chemical conditions used in the main experiments (the different pH and GSH concentrations). Without this, it is believable but not conclusive that induced changes in T1 in the cleavable nanoparticles are entirely due to the desorption of Gd⁺. There could be an effect on what the authors call "intrinsic relaxation rate", and this can be verified in the proposed control experiment.

In the chirp sensitivity enhancement plot in Fig.3b, how far is the reference square pulse from optimal pulse parameters (frequency, duration and microwave power)? If these parameters are

optimized, there should not be any enhancement, and in fact, the chirp scheme would probably perform worse because of the finite sweep rate (this non-adiabaticity should be also be discussed). In that sense, I am not sure what this plot represents. There must be a better way to show that the chirp technique is robust against parameter changes, while the square pulse technique is less robust (and that they converge for a narrow range of parameters).

Fig.4b shows the maximal achievable contrast in T1 to be between 10us and 60us for the nanodiamonds with and without Gd+, respectively. However, in Fig.4c , pH-induced desorption of Gd+ makes T1 rise and saturate near 30us. Based on the model in Fig.5, this corresponds to a maximum reduction in surface Gd+ concentration by a factor of few only. What contributes to this reduced contrast? Why does such a large Gd+ fraction remain?

Minor Comments:

What is the significance of doing these experiments in a microfluidic channel?

It would be nice to overlay the ICP MS release curves and T1 increase curves on the same plot, and also to have error bars on the ICP MS data.

For non-experts, it would be helpful to explain more clearly why the NV sublevel population needs to be flipped before readout for a more accurate T1 measurement.

[1] Kaufmann et. al., Detection of atomic spin labels in a lipid bilayer using a single-spin nanodiamond probe. PNAS, 110, 10894–10898 (2013)

[2] Sushkov et. al., All-optical sensing of a single-molecule electron spin. Nano Letters, 2014, 14, 6443–6448 (2014)

Reviewer #3 (Remarks to the Author):

Summary of the key results: the paper describes a new method for sensing biological environments based on T1 relaxation measurement in nanodiamonds which are coated with polymer coating with Gd based relaxation centers bound to it. The detachment of the Gd from the nanodiamond as a result, for example, of changing pH results in detectable change in T1.

Originality and interest: this research is novel and original.

Data & methodology: validity of approach, quality of data, quality of presentation: The paper is clearly written; the text and illustrations are nice and informative.

Appropriate use of statistics and treatment of uncertainties: everything seem good with one exception. The error bars in Figs. 4 c,d and S4 b appear to be significantly larger than the spread of the points. This would mean an unrealistically low χ^2 . The authors should check the analysis, for example, might it be that they draw error bars corresponding to the standard deviation (incorrect) vs. standard errors (correct)?

Conclusions: robustness, validity, reliability: the conclusions are robust and reliable.

Suggested improvements: experiments, data for possible revision: no obvious problems or weaknesses.

References: appropriate credit to previous work? Yes

Clarity and context: lucidity of abstract/summary, appropriateness of abstract, introduction and conclusions: everything is in order.

Small things:

* Extra "in" in line 171

* In the same section, one needs to look into Supplementary to understand what the "sensitivity enhancement" is. Maybe better to explain this in the main text?

Also:

What are typical laser powers used to pump and read-out the spin-states of the NV centers during the measurements? It would be interesting if the authors could discuss the effects of the pump laser on the polymer coating-bearing spin labels, and if the pump beam can result in release of Gd³⁺ ions from the polymer coating/surface.

The single tau measurements assume that there is no loss of contrast (which is proportional to the measurement at $F(\tau_1)$ - see line 343). Is there any reason why this should be always the case?

Line 276: Typo "Handbury-Brown and Twiss"

Line 143: I'm not certain I understand why this measurement is called an in-situ measurement when one has to incubate, centrifuge, and then remove the supernatant liquid for a measurement - and repeat that for different incubation times to obtain temporal information.

Dear Dr. Matsiko,

We appreciate the great efforts and valuable time you and the reviewers have taken to comment on our work (original Manuscript ID: NCOMMS-16-17056). We thank all the reviewers for acknowledging our work is novel. We are grateful for the reviewer #2 giving us several detailed comments about the required control experiments, and have addressed all of the raised questions.

According to the reviewer's comments, we have extensively revised the paper and added new data as suggested. Perhaps we should apologize that we have spent almost 3 months to do the revision since we have done a significant amount of additional experimental work and clarification including preparing new samples, performing new measurements and evaluating them. We believe that the quality of current paper has been greatly improved after revision.

We look forward to your favorable considerations.

Best regards

Zhiqin Chu and Petr Cigler

Reviewers' comments:

Reviewer #1 (Remarks to the Author):

1 The only concern that I have is that the changes in pH that were considered are quite drastic. So I would like the authors to comment on the resolution in pH/redox potential they can actually resolve. How much difference in pH/redox potential can you detect? However, since this application is so new and this is the first proof of principle data I think even if the sensors might not yet be fit to detect biologically relevant changes this is still a milestone in that direction. Overall this article is very good and I recommend to publish it (almost) as it is.

Response to reviewer:

We are grateful for reviewer considering the overall quality of our work is very good and recommendation for publish as it is. To clarify the only question raised here, we performed the time dependent ensemble T_1 measurement with ND@pH nanosensor in buffers covering the physiologically relevant pH range (pH=3.8, 4.5, 5.5, 6.2, 6.9). Since our measurement of pH is based on changes of T_1 , we choose the data points within the first 12 minutes and performed a linear fit for extracting T_1 changing rate (Fig. R1). As shown in Fig. R1, it is quite clear that the extracted T_1 changing rate at different pH value is significantly distinguishable. This indicates that our NV relaxometry based method (T_1 measurement) can at least resolve a pH difference ~ 0.7 within pH range 3.7-6.9. We present these results in the manuscript and show these graphs in revised Supplementary Information (Fig. S8).

Fig. R1. Dependence of the fitted T_1 changing rate of ND@pH nanosensor on pH. Six T_1 measurement points (120 seconds per point) were linearly fitted to extract the actual T_1 change rate corresponding to a particular pH. Britton-Robinson buffers with same composition were used. Their ionic strength was normalized using KCl to ensure isoosmolar environment for measurements.

Reviewer #2 (Remarks to the Author):

1 This method of magnetic noise detection from Gd⁺ (and other targets) via NV centers in diamond (including nanodiamonds) is well established (as in references 34,35,44 cited by the authors and [1,2] below which should also be cited). This type of co-polymer functionalization of nanodiamonds has also been shown before by multiple authors (including some of the current authors). The novelty in this work lies in combining the NV/Gd⁺ relaxometry technique with nanodiamond functionalization unstable under certain chemical environments and thereby demonstrating the detection of the said chemical environments.

Response to reviewer:

We thank reviewer for acknowledging the novelty in current work. However, the novelty is in our view not only limited to the mentioned “combining the NV/Gd³⁺ relaxometry technique with

nanodiamond functionalization". In addition, we believe the current work has the following features also contributing to the novelty:

1) The adopted linear chirp pulse replacing square pulse enables a more robust way to perform T_1 NV relaxometry measurement. To the best of our knowledge, this has never been achieved before in the field. In our view the robust measurement scheme are the key ingredient in successful adapting quantum sensing algorithms to nanodiamonds in a complex environment like cells. Especially when measuring multiple nanodiamonds (with ensemble NV centers) that differ in their spin signature with the highest performance, i.e., nanodiamonds are localized in one organelle or form clusters upon cellular uptake, it is quite difficult to perform T_1 measurement with square pulse.

2) The involved precise chemistry and architecture of the polymer for nanodiamond surface functionalization. In particular, we show the polymer shell does not swell/shrink under different pH/ion concentrations, which is a kind of innovation among polymer coatings. In addition, we achieved very high degree of polymer substitution (~30 molar %) compared to other works (typically 5-12%) [Nanoscale 2015, 7, 415; Biomaterials 2014, 35, 5393; ChemPlusChem 2014, 79, 21] while keeping an excellent colloidal stability of the nanoparticles.

3) The novel chemical strategy for synthesis of Gd^{3+} -complexes bearing multipurpose linkers. We made available chemical functionalization of highly stable Gd^{3+} -complexes for click chemistry and demonstrated a general and convenient pathway to Gd^{3+} -complexes detachable *via* cleavable linkers. We foresee these types of complexes can be used in attractive applications such as chemically selective MRI imaging. To make this contribution and reaction pathways clearer, we added to revised Supplementary Information two new schemes (Scheme S1 and S2) explaining the new synthetic pathways in more detailed way. We also added a sentence to the revised manuscript (Page 5, line 6-8) referring to these schemes. We have also cited the additional reference as suggested by reviewer, see Page 4, line 25 in the revised manuscript.

Scheme S1: Synthesis of redox sensitive complex $[Gd(L^1)]^-SS-N_3$

Reaction conditions and yields: i) Hexamethyldisilazane, under Ar, 105 °C, 12 h ii) 3-Bromo-1-(trimethylsilyl)-1-propyne (**1**)/CH₂Cl₂, under Ar, -10 °C to rt, 24 h, 54 % iii) methyl chloroformate, pyridine/CH₂Cl₂, reflux, 15 min, 97 % iv) *t*Bu₃DO3A·HBr (**4**·HBr), (CH₂O)_{*n*}/MeCN (dry), 65 °C, 3 days, ~90 % v) 1. 85% aq. HCOOH, 65 °C, 4 days, 2. 1,5% aq. HCl, rt, overnight, 78% vi) GdCl₃·5H₂O/aq. NH₄OH, pH 4,8, rt, overnight, ~60 % vii) 1,6-bis(azido)-3,4-dithiahexane (**6**), CuSO₄, NaF, Sodium Ascorbate/ THF/PrOH:H₂O (1:1:2), rt, 12 h, ~50%

Scheme S2: Synthesis of pH sensitive complex $[\text{Gd}(\text{L}^2)]^- - \text{NHNH}_2 - \text{N}_3$

Reaction conditions and yields: i) 1. $\text{NaNO}_2/\text{aq. HCl}$ 0 °C, 25 min; 2. $\text{NaN}_3/\text{aq. HCl}$ 0 °C, 25 min, 70 % ii) $\text{GdCl}_3 \cdot 5\text{H}_2\text{O}/\text{aq. NH}_4\text{OH}$, pH 9, rt, overnight, ~70 % iii) pent-4-ynehydrazide (**8**), CuSO_4 , Sodium Ascorbate/ $\text{THF}:\text{H}_2\text{O}$ (1:1), rt, overnight, ~60 % iv) 6-azido-hexan-2-one (**9**), AcOH , $\text{MgSO}_4/\text{MeOH}$ (dry), reflux, 3 days, ~30 %

Major Comments:

2 The system is out-of-equilibrium and the sensor operation is non-reversible. Namely, in the case of ND@pH , the polymer to Gd^{3+} complex link is unstable on the time scale of hours even in neutral water, based on the data in the paper (Figures 2 and 4). Reduced pH simply accelerates the Gd^{3+} desorption by a factor of few, depending on the pH. The same applies to increased GSH concentration. In other words, the authors don't actually measure pH or redox potential, but polymer cleavage rates, which are affected by these variables. This raises several concerns:

Response to reviewer:

We agree with reviewer that we are actually measuring the change of T_1 given by local pH or GSH concentration and the current designed system is non-reversible. However, the “factor of few” is not suitable for describing the changes in Gd^{3+} -complexes desorption rate. For example, T_1 slope (which corresponds to the desorption rate) changes 22x upon change from pH 7.4 to 4.5 (what corresponds to change from extracellular environment to lysosome). The maximal observed change is ~50x (pH 7.4 to 2.0). To investigate the stability of pH sensitive ND@pH sample, we measured the release of Gd^{3+} complex from such sample at pH 7.4 using ICP MS (insert in Fig. R2). The ND@pH particles easily reach plateau at pH 2.0 and pH 4.5 within 20 minutes, however, it takes roughly 800 minutes for pH 7.4. This is almost ~40x difference in the measured effective releasing time window. For the current study, the chosen measurement time was around 1 hour. Obviously, the pH sensitive ND@pH nanosensor is stable within such time period. We have added corresponding description in revised manuscript (Page 6, line 23-27; page 7, line 1-5). Moreover, we show in Fig. R1 (stimulated by a comment of Reviewer #1) that in the physiologically relevant pH range we can estimate a pH value within 12 min with precision at least 0.7 pH unit.

Fig. R2. The Release kinetics of Gd³⁺ complex in pH sensitive ND@pH particles in pH 2.0, 4.5, and 7.4 buffers analyzed by ICP MS.

2a. 1) Only unidirectional changes in these variables can be detected (decrease in pH or increase in GSH).

Response to reviewer:

We agree with reviewer that our current sensor is not working in reversible way at the moment. Nevertheless, we want to emphasize that the aim of our current study is not to demonstrate that we have designed the best hybrid sensor system. In fact, the current work is more about exploring the potential of using the nitrogen vacancy centers embedded in nanodiamonds for quantitative sensing of chemical reactions. In principle, the nitrogen vacancy centers allow to readout the state of surrounding spin labels via a robust and well established method, namely NV spin relaxometry [PNAS 2013, 27, 10894-10898; Nat. Com. 2013, 4, 1607]. In this regard, the challenge of constructing current hybrid sensor is shifted to design a chemically programmable interfacial layer on the diamond lattice containing cleavable spin labels. In particular, our designed polymer is insensitive to the change of local chemical environments (e.g., pH and ionic strength), which is critical for reliable and robust measurement based on NV spin relaxometry. One should notice that the current demonstrated detection method has a promising potential for getting access to more other sensing options, either in an irreversible or even a reversible way [Mater. Sci. Eng. R. Rep. 2015, 93, 1-49] with proper chemical engineering of the attached polymer.

For example, we are currently developing a crosslinked multicomponent co-polymer system on nanodiamonds which changes its overall geometry upon chemical stimuli originating in close environment of the particles. This system responds in a completely reversible manner. We believe this approach can overcome the disadvantage of unidirectionality commented by Reviewer #2. To exemplify our achievements, we show here preliminary results from DLS measurements of our first generation of pH-responsive polymer shell synthesized on NDs (Fig. R3). We also added a sentence to the manuscript showing limitation of our approach and the possible solution of it in this context: "To enlarge the measurement window to basically unlimited time, we are currently developing a reversibly responding polymer coating on NDs which operates without a need of irreversible cleavage events." See Page 12, line 5-7 in the revised manuscript.

Fig. R3. Nanodiamonds coated with a crosslinked pH-responsive co-polymer system respond reversibly to different pH values by swelling/contracting of the polymeric shell. The particles show monotonic increase in their hydrodynamic diameter (measured using dynamic light scattering) with exception of pH value <5, where the particles are highly protonated, decharged, and show a tendency to aggregate (left). Scheme of the particles structure and a simplified mechanism of reversible charging/decharging is shown on right. Currently, we are working on optimization of this system and extending its colloidal stability to a wider range of pH.

2b) 2) There is not a one-to-one mapping between these chemical variables and the detected T1 signal; the signal depends on absolute time since the introduction of the sensor into the system and on the history of changes in these chemical variables.

Response to reviewer:

We agree with reviewer that the signal obtained from the current sensor depends on the time after adding into the measured system. However, as we discussed before, this is part of the intended sensor design depending on the specific application. As long as the sensor has not released all the bound Gd^{3+} -complexes, it should be possible to estimate its release rate which is determined by local chemical environments (e.g., pH values). In a certain time window the release rate can be approximated by a linear fit function. To demonstrate this in a more detailed way we performed an additional experiment using a buffer set with more subtle differences between individual points (Fig. R1). We found the estimated release rates depend monotonously on pH (Fig. R1). These measurements show that the sensor can operate in predictable mode at least within a certain time window. We added the following sentences to the manuscript: “Using set of Britton-Robinson buffers (with equal composition and ionic strength) we also investigated the accuracy of our sensor in the physiologically relevant pH range (3.7–6.9) for a short measurement time frame (12 min). We estimated the slopes of T_1 change for each measured pH and found their monotonous dependence on pH (Supplementary Fig. S8) allowing for accuracy of a pH difference at least ~ 0.7 .” See Page 9, line 1-5 in the revised manuscript.

2c) 3) The response of the signal to a fast change in pH or GSH is on a time scale of tens of minutes, which is too slow for many biological processes on the microscale and nanoscale.

Response to reviewer:

The sensor responds, in fact, to a change in pH or GSH, immediately, by a change in the release rate of Gd^{3+} -complexes. The measurement time is therefore determined by the time needed for estimating the corresponding release rate which is typically 12 min or less with the full T_1 curve measurement approach (as we documented in Fig. R1). However, one should notice that the requirement for measurement time is varying in different applications. For example, endosome maturation process inside cells will normally take hours to develop, i.e., from early endosomes to late endosomes/lysosomes [Nat. Rev. Mol. Cell Biol. 10, 597–608 (2009)]. In this sense, our current sensor is suitable for monitoring corresponding processes—for example, the pH sensor should be

suitable for observing the process of endosome maturation (A method with minutes scale temporal resolution should be sufficient to monitor an event with hours scale).

As a proof-of-concept demonstration, we developed here the pH and GSH sensitive samples which work in minutes scale with full T_1 curve measurement. In fact, the measurement time can be reduced to within tens of seconds through fixed tau measurement (as demonstrated in Fig. 4e). To further increase the temporal resolution of our sensor, we are also investigating on cleavable bonds with much faster reaction rates, i.e., peptide linkers selective to certain intracellular proteases. We believe this will push down the temporal resolution of our sensor from minutes to tens of seconds in the near future.

2d. 4) As equilibrium is approached (the nanodiamonds loose all Gd+) on the hour-long time scale, the sensor stops working, which limits the length of time over which a system of interest can be monitored.

Response to reviewer:

We agree with reviewer that the sensor stops working as equilibrium is approached. However, for pH values higher 7, the time necessary for reaching such equilibrium is relatively long (see the release curve in Fig. R2, inset). For example, at pH 7.4 (typical for cell media) it would remain functional for many hours.

Notably, many biologically important processes are off-on (or on-off) type: we do not need to quantify, just detect that a biological/chemical event occurs. In such cases, one can imagine construction of a sensor based on the same principle as our pH and redox sensor (i. e., release of a paramagnetic complex from nanodiamonds proximity), but with a very stable linker with highly specific cleavage site. For example, we are currently working on the development of stable peptide linkers accessing highly selective detection of proteases appearing immediately in cytosol after blast of oxidative stress (off-on event). There are also further options for sensing architectures, as we suggested in our response to Reviewer 2, comment 2a.

2e. 5) Since the authors present this as a sensing method, plots summarizing its sensitivity is in order, such as plots of time constant of T_1 increase versus pH and GSH concentration, and perhaps plots of the slopes of the previous plots versus pH and GSH. How does this performance compare to other sensing modalities of pH and GSH in biological systems? What are the typical magnitudes of the changes in pH and GSH in biological systems, at the relevant pH and GSH values, and can this technique realistically be used to measure these changes (in terms of sensitivity - besides the time scale and irreversibility problems mentioned above)?

Response to reviewer:

As shown in Fig. R1, we can at least resolve a pH difference of 0.7. Compared to other detection mechanisms achieving resolution approximately ± 0.1 pH unit for intracellular measurement (for example: Analyst, 2013, 138, 3126), our resolution is lower. On the other hand, some optical pH sensors exhibit a sigmoidal response toward changes in pH, and the dynamic range is usually limited to ca. $pK_a \pm 1.5$ [Anal. Chem., 2014, 86, 15]. Our system operates in quite a broad pH range. Considering practical measurements in cells, the accuracy of our sensor is sufficient, as differences between extracellular space, cytosol and some organelles are much higher than 0.7. For example, the cytosol pH is between ~ 7.4 while endo/lysosomal compartments show pH ~ 4.5 [Nat. Rev. Mol. Cell Biol. 10, 597–608 (2009)]. The intracellular GSH concentrations usually range from 0.5 to 10 mM, whereas extracellular values are almost three orders of magnitude lower [J. Amino Acids 2012, 2012, 736837]. This difference lies again in a range well measurable by our nanosensor.

Besides this argumentation, we would like to emphasize that we did not either aim or claim to construct a best (most sensitive and/or fastest) pH or redox nanosensor. Our goal was rather to introduce a broader concept demonstrating that chemical reactions can be monitored using NV centers relaxometry which we exemplified using two types of nanosensors with conceptually new architecture and way of detection.

Irreversibility (and unidirectionality): The vast majority of available chemical or nanoparticulate probes for glutathione imaging are based on irreversible mechanisms of detection (for review see Chem. Soc. Rev. 2013, 42, 6019; J. Cell. Biochem. 2014, 115, 1007). Moreover, a surprisingly high number of currently used nanosensors is based on mechanisms which are either irreversible (based on formation or cleavage of covalent bonds) or irreversible in principle, because the formed non-covalent sensing assembly is extremely stable (for instance, nucleic acid hybridization, antibody and aptamer affinity probes, FRET sensors utilizing cleavage reactions; for overview, see Science 346, 2014, 1247390-1).

To clarify our goals and to improve discussion in the manuscript, we added comments reflecting this reasoning to the revised manuscript (Page 10, line 17-28; Page 11, line 1-12).

3. In view of comments 1-3 above, it is not very fair to call this technique time-resolved detection of chemical signals. Perhaps the authors can discuss situations where it would be fair and where these issues are mitigated. Alternatively, what are ways to mitigate them in future experiments?

Response to reviewer:

We agree with reviewer that our technique does not allow for a “real-time” resolved detection of chemical events. We have corrected in the title into “Optical imaging of localized chemical events using programmable diamond quantum nanosensors” accordingly. To achieve the goal of detection in a real-time manner, our next step is to work for polymer chemistry with full reversibility and reduce the measurement time for improved temporal resolution (also see our response to Reviewer #2, comment 2a). However, we should emphasize that the phrase “real-time” actually has a relative meaning when considering the specific application. For example, our current nanosensors can work in a “real-time” manner for monitoring pH changes associated with the endosome maturation (also see our response to Reviewer #2, comment 2c).

4. The imaging part of the experiment, T1 was imaged over a surface covered with two types of nanodiamonds that underwent different acid treatment and hence had different Gd+ surface concentrations. Different T1 times were optically resolved on different nanodiamonds. A more thorough discussion is needed here. Namely, I have two concerns:

4a. 1) On the T1 image in Fig.4, only those nanodiamonds were highlighted where T1 times were long and the nanodiamonds were incubated in pH=2, or where T1 times were short and the nanodiamonds were incubated in pH=7. Clearly however, several nanodiamonds in that image do not show this correlation. To make clear statement about these results, a summary plot is in order, such as overlapped histograms of T1 times for the two types of nanodiamonds. What explains the apparently large variance?

Response to reviewer:

We thank reviewer for pointing this out and we have added the statistical view of T₁ observations for individual ND@pH nanosensors adsorbed on coverglass in a microfluidic chamber at different pH buffers (pH 2.0 and 7.4) as shown in Fig. R4. The observed variance (individual nanodiamonds spots) between different nanodiamonds mainly comes from nanodiamonds themselves,

this is also the reason why we performed most of the measurements through ensemble manner (see details in Methods in manuscript), representing the average value of the chosen sample batch. To easily distinguish the T_1 mapping, we now modified the color scale of the image, see revised Fig. 4g. We have revised/added corresponding figures/descriptions in revised manuscript (Page 9, line 25-27) and Supplementary Information (Fig. S9)

Fig. R4. Statistical view of ND@pH nanosensors in pH 7.4 and pH 2.0 buffers. After injecting the sample into a microfluidic chamber, the individual relaxation times for different adsorbed ND@pH nanosensors had been measured at pH 7.4 (green) and after changing to pH 2 (red).

4b. 2) Since the imaging was a static experiment performed after separate chemical treatments of different parts of the sample, it is unfair to make the claim of "real-time imaging of localized chemical events" in the paper and especially the title.

Response to reviewer:

We agree with reviewer that calling our technique "time-resolved" might be misleading. We thank for pointing this fact out. We have corrected the title into "Optical imaging of localized chemical events using programmable diamond quantum nanosensors" accordingly (also see our response to Reviewer 2, comment 3). Furthermore, we have removed all the description with "real-time" in the revised manuscript.

5. An important control experiment is missing: T_1 measurements of non-cleavable nanoparticles ND-HPMA and ND-NPMA-Gd³⁺ under the different chemical conditions used in the main experiments (the different pH and GSH concentrations). Without this, it is believable but not conclusive that induced changes in T_1 in the cleavable nanoparticles are entirely due to the desorption of Gd³⁺. There could be an effect on what the authors call "intrinsic relaxation rate", and this can be verified in the proposed control experiment.

Response to reviewer:

We agree with reviewer that the T_1 measurements on control sample should also be shown. We have now added a systematic study (T_1 measurement) with control samples including both ND-HPMA and non-cleavable ND-HPMA-Gd³⁺ particles in different pH (2.0, 4.5, 6.0 and 7.4) and GSH (1, 5 and 10 mM) buffer conditions. As shown in Fig. R5 and Fig. R6 below, it is quite clear that there is no apparent Gd³⁺ release from the control samples checked either by ICP-MS and our T_1 measurement. We have added the corresponding data to revised Supplementary Information (Fig. S4 and S5).

Fig. R5 The control sample in different pH buffers checked by ICP-MS and T_1 measurement. Release kinetics of Gd^{3+} -complex from ND-HPMA-Gd (non-cleavable) checked by ICP MS (upper panel) in different buffers (pH 2.0, 4.5, 6.0, and 7.4). Time-dependent ensemble measurement for T_1 (lower panel) of ND-HPMA and ND-HPMA-Gd (non-cleavable) in different buffers (pH 2.0, 4.5, 6.0, and 7.4). The slope of all linear fit is approaching zero, indicating little change of Gd^{3+} -complex within the measurement time period.

Fig. R6 The control sample in GSH buffers with different concentrations checked by ICP-MS and T_1 measurement. Release kinetics of Gd^{3+} -complex from ND-HPMA-Gd (non-cleavable) checked by ICP MS (upper panel) in different GSH buffers (1 mM, 5 mM and 10 mM GSH). Time-dependent ensemble measurement for T_1 (lower panel) of ND-HPMA and ND-HPMA-Gd (non-cleavable) in different GSH buffers (1 mM, 5 mM and 10 mM). The slope of all linear fit is approaching zero, indicating little change of Gd^{3+} -complex within the measurement time period.

6. In the chirp sensitivity enhancement plot in Fig.3b, how far is the reference square pulse from optimal pulse parameters (frequency, duration and microwave power)? If these parameters are optimized, there should not be any enhancement, and in fact, the chirp scheme would probably perform worse because of the finite sweep rate (this non-adiabaticity should be also be discussed). In that sense, I am not sure what this plot represents. There

must be a better way to show that the chirp technique is robust against parameter changes, while the square pulse technique is less robust (and that they converge for a narrow range of parameters).

Response to reviewer:

We thank reviewer for pointing out that we should clearly explain the advantages of chirp techniques for T_1 measurement. Therefore, we revise the corresponding descriptions/figures and add more supporting data as following. For all ensemble T_1 measurements in the current study, we are actually measuring ensemble NV centers (several different nanodiamond particles with random orientation and slightly variations in their resonance lines) rather than a single NV center in a fixed nanodiamond. This is especially applicable to the case that nanodiamond particles were feeding to cells where different nanodiamond particles were randomly clustered into aggregates inside cells [Scientific Reports 4, 2014, 4495]. In this regard, we want to emphasize that our work adopts a single-step method providing strategies for a robust sensing scheme without required preliminary measurements which is necessary when using square pulse.

To compare the performance of different schemes used in T_1 measurement, we extracted the spin contrast from experimental measurement as a function of driving strength (estimated by Rabi oscillations on the left ODMR transitions as shown in Fig. R8a) for the control sample ND-HPMA (without Gd^{3+} complex) (Fig. R7b). The obtained spin contrast decreased as the driving strength reduced for both measurement scheme, but the chirp pulse almost always resulted in a higher contrast than that obtained with square pulse. We also plotted the extracted contrast as a function of expectable minimal adiabaticity factor Q (a measure of the adiabaticity of the used pulse scheme [Journal of Magnetic Resonance 2001, 153, 155-177]) when using linear chirp pulses at different sweep rates (Fig. R7c). For almost all the measurements using linear chirp pulse, we experimentally obtained a contrast ranging from 4% to 10 %, corresponding to a factor Q in the range of 3 up to more than 10. This indicates that we are working in the adiabatic regime. In the current study, we actually utilize the adiabatic passage in the form of a linear chirp pulse to enhance the NV spin sublevel flip. As the NV sublevel flip determines the contrast in T_1 measurements, the linear chirp pulse will lead to a better spin contrast.

To better understand the observed phenomenon, we simulate the NV spin state evolution excited by different pulse schemes in the presence of inhomogeneous ODMR line broadening (a few MHz as shown in Fig. R8a), especially for the nanodiamonds (details in revised SI). One should notice that the NV spin flip probability is proportional to the contrast for relaxation measurements, the higher the flip probability the higher the contrast. As shown in Fig. R8b&c, the simulated NV sublevel flip probability with different ODMR line broadening using different pulse scheme is plotted as a function of the driving strength. If no inhomogeneous broadening is introduced, the performance of the square pulse is independent of microwave power (Fig. R8c, red line). However, the more the broadening is the more power is needed to efficiently flip the NV sublevel population when a square pulse is used (Fig. R8c blue line). As a comparison, the linear chirp pulse is totally insensitive to the ODMR line broadening in the range of simulated parameters (Fig. R8b). In fact, the chirp pulse resulted in at least a two times sensitivity enhancement with different chirp rates when the microwave power is large enough (Fig. R9). In summary, the linear chirp pulse is more robust against changes in microwave excitation and can drive NV spins in a broader frequency band. We have added corresponding figure in revised manuscript (Fig. 3); descriptions in the revised manuscript (Page 7, line 19-27; Page 8, line 1-10; Page 12, line 18-21); and figure in revised Supplementary Information (Fig. S6 and S10).

Fig. R7 Robust relaxation measurement with linear chirp pulse. (a) Schematic cartoon showing the used optical and microwave pulse sequence for T_1 relaxation measurement. (b) Comparison of experimentally extracted contrast of T_1 relaxation measurement with chirp pulse and square pulse for control sample ND-HPMA; The contrast is defined as the normalized initial difference between the sensing sequence with and without the inversion pulse. (c) Dependence of extracted T_1 contrast on the extracted minimum adiabaticity factor Q , where Q is defined as the effective driving amplitude over its angular velocity. The error bars represent the 95% confidence intervals.

Fig. R8 Simulation of the NV sublevel flip excited by different pulse scheme. (a) A typical measured ODMR line for a randomly picked cluster of the ND-HPMA particles (black) excited with $45\mu\text{W}$ laser and weak microwave. The ODMR spectrum had been fitted by two Voigt profiles with a fixed Lorentzian linewidth of 1 MHz and a variable Gaussian linewidth (see revised SI for used parameters). Both transitions in the measured ODMR line are quite broad and separated by around 16 MHz. (b) Simulated NV sublevel flip probability excited with a linear chirp pulse and square pulse plotted as a function of microwave driving strengths. The probability was calculated for an ensemble by averaging over all polar angle and assuming an inhomogeneous broadening of both ESR transitions with a variable value (color coded from 0 MHz (red) to 5 MHz (blue)). Both transitions of ODMR line are set to split in average ~ 16 MHz to mimic the experimental situation in (a). The parameters of the chirp pulse had been set to typical values used in our real measurement (50 MHz sweep bandwidth with a 20 kHz/ns sweep speed).

Fig. R9 Relative enhancement for chirp pulse versus a square pulse. (a) Comparison of experimentally obtained sensitivity between T_1 relaxation measurement with linear chirp pulse over square pulse; the sensitivity enhancement is calculated as the ratio of sensitivity obtained with linear chirp pulse to that with square pulse. (b) Simulated relative enhancement of the probability to depopulate the NV $m_s=0$ sublevel after applying linear chirp pulse over square pulse with different microwave driving strength. The error bars represent the 95% confidence intervals.

7. Fig.4b shows the maximal achievable contrast in T_1 to be between 10 μ s and 60 μ s for the nanodiamonds with and without Gd³⁺, respectively. However, in Fig.4c, pH-induced desorption of Gd³⁺ makes T_1 rise and saturate near 30 μ s. Based on the model in Fig.5, this corresponds to a maximum reduction in surface Gd³⁺ concentration by a factor of few only. What contributes to this reduced contrast? Why does such a large Gd³⁺ fraction remain?

Response to reviewer:

The sample shown in Fig.4b is actually a pure control samples: ND-HPMA (60 μ s) and non-cleavable ND-HPMA-Gd (10 μ s), while the sample shown in Fig.4c is the pH sensitive samples. These are totally different batches of sample. We attribute this difference to the various batches sample with slightly different loading of Gd³⁺ ions.

Based on the model in Fig.5 in the main text, the different T_1 value is actually corresponding to a calculated concentration of Gd³⁺ complex ~ 0.02 M ($T_1 = \sim 35 \mu$ s, pH sensitive sample) and 0.002 M ($T_1 = \sim 60 \mu$ s, pure control sample). Considering the initial loading concentration of Gd³⁺ complex ~ 0.2 M, the release ratio $((C_{\text{initial}} - C_{\text{final}}) / C_{\text{initial}})$ is 90% for $T_1 = 35 \mu$ s and 99% for $T_1 = 60 \mu$ s, respectively. It is not surprising that the real case (pH sensitive sample) differs (9%) from the extreme case (without Gd³⁺ complex).

8. Minor Comments:

8a. What is the significance of doing these experiments in a microfluidic channel?

Response to reviewer:

Since we need to change the solution for changing pH or GSH in a well-controlled manner (especially necessary for Fig. 4 f&g), we adopted the microfluidics devices.

8b. It would be nice to overlay the ICP MS release curves and T_1 increase curves on the same plot, and also to have error bars on the ICP MS data.

Response to reviewer:

We put the systematic studies on control samples through both T_1 and ICP MS measurements on the same plot, as show in Fig. R5 and Fig. R6. These Figures have been added to revised Supplementary Information, see Fig. S4 and S5.

8c. For non-experts, it would be helpful to explain more clearly why the NV sublevel population needs to be flipped before readout for a more accurate T_1 measurement.

Response to reviewer:

We thank reviewer for pointing this out, and have added an additional explanation as following. The charge state of the NV centers can be influenced by various quantities, such as the local environments, surface treatment of the diamond, excitation wavelength of laser and etc. [Appl. Phys. B 2006, 82, 243; Phys. Rev. B 2011, 83, 081304; Adv. Funct. Mat. 2012, 22, 812-819; New J. Phys. 15, 013064]. In particular, the charge state of NV centers in nanodiamonds is highly depending on the surface passivation due to their high surface to volume ratio [Adv. Funct. Mat. 2012, 22, 812-819]. In addition, their charge state can adjust over time without illumination and also differs under laser illumination. As a consequence, the nitrogen vacancy centers can change their charge state on the time scale from several μ s to ms, depending on the used laser power and wavelength, which can be directly observed as an increase or decrease of the NV fluorescence after turning on the laser [Appl. Phys. B 2006, 82, 243; New J. Phys. 15, 013064]. A typical single T_1 measurement containing no additional control sequence will contain that information, diminishing the measured spin contrast significantly. If one now applies a second measurement (control), i.e., by an additional microwave pulse (e.g., square or linear chirp pulse) before read out, the NV sublevel population is inverted and the pristine spin contrast can be obtained. In other words, one can extract the pristine spin contrast by subtracting the normalized T_1 measurement from the control [Nat. Nano. 2015, 10, 125-128]. We have added the corresponding description in the revised Supplementary Information (Supplementary text) and mentioned them in revised manuscript (see Page 7, line 19).

Reviewer #3 (Remarks to the Author):

1. Small things:

1a. * Extra "in" in line 171

Response to reviewer:

We have deleted the extra "in" pointed by reviewer in the revised manuscript

1b. * In the same section, one needs to look into Supplementary to understand what the "sensitivity enhancement" is. Maybe better to explain this in the main text?

Response to reviewer:

We have added the description in the corresponding main text, see page 7, line 19-27; Page 8, line 1-10 in the revised manuscript (Please also see our response to reviewer 2, comment 6.)

1c. Also: What are typical laser powers used to pump and read-out the spin-states of the NV centers during the measurements? It would be interesting if the authors could discuss the effects of the pump laser on the polymer coating-bearing spin labels, and if the pump beam can result in release of Gd^{3+} ions from the polymer coating/surface.

Response to reviewer:

We agree with the reviewer that it is reasonable to check the effects of laser on the release of Gd^{3+} complex from the polymer shell. To investigate the exact influence of laser power, we performed the ensemble T_1 measurement with the non-cleavable ND-HPMA-Gd particles (as used in Fig. 4 in the

revised main text) under different laser power ranging from 500 μW to 16mW (Fig. R10). In fact, this range covers all the used laser power for ensemble measurements in current study (e.g., 4-8 mW). It is quite clear that there is no difference between measurements among different laser power, indicating the used laser power has little effect on Gd^{3+} complex release.

Fig. R10. T_1 measurements as a function of excitation laser power on non-cleavable ND-HPMA-Gd particles. Right panel shows the sum of all measured data points for a certain laser power shown in left panel (laser power is changing as a function of time). The error bars (right panel) are calculated by a 2*sigma confidence interval. The blue area marks the range of laser power used for all the ensemble measurements in current study. The red dashed line marks the average T_1 value $\sim 11 \mu\text{s}$ measured at different excitation laser power.

1d. The single tau measurements assume that there is no loss of contrast (which is proportional to the measurement at $F(\tau_1)$ - see line 343). Is there any reason why this should be always the case?

Response to reviewer:

The single tau measurement only reduces the sample points into two but without changing any measurement parameters (microwave, period, and etc.). In principle one reference the spin contrast at a certain waiting time τ to the initial spin contrast. Here we actually did four individual measurements: Two measurements with and without a spin inversion at $\tau = 0$ (reference) and another two at a τ of our interest (sensing). Even the overall contrast of the particles changes, the measured fraction of the extracted spin contrast remains constant (sensing divided by reference) [Nat. Nano. 2015, 10, 125-128]. But as the reviewer is implying the single tau measurement has a certain operational τ window (in terms of sensitivity) as can be seen when calculating the SNR of a single tau measurement [10.1103/PhysRevB.87.235436]:

$$SNR \propto \sqrt{\tau} \cdot e^{-\frac{\tau}{T_1}}$$

The maximum SNR is reached when $\tau = \frac{T_1}{2}$.

1e. Line 276: Typo "Handbury-Brown and Twiss"

Response to reviewer:

We have corrected the typo "Handbury" to the corrected "Hanbury" in the revised manuscript (Page 13, line 6).

If Line 143: I'm not certain I understand why this measurement is called an in-situ measurement when one has to incubate, centrifuge, and then remove the supernatant liquid for a measurement – and repeat that for different incubation times to obtain temporal information.

Response to reviewer:

We agree with reviewer and have deleted the phrase “in-situ” for precise expression in the revised manuscript.

List of changes:

Page 2, line 7

We have corrected the sentence of the abstract to “We adopt a single-step method to measure spin relaxation rates enabling time dependent ...”.

Page 4, line 8-9

We have corrected the sentence to “We designed and demonstrated time dependent ...”.

Page 4, line 25

We have added two additional reference suggested by reviewer #2.

Page 5, line 6-8

We have added two new Schemes (see Schemes S1 and S2 in Supplementary Information) and rewritten the sentence here as “We developed a convenient synthetic pathway to their modification with cleavable...”.

Page 6, line 2-3

We have rewritten the sentence as “with square root of the Gd^{3+} concentration in the shell...”.

Page 6, line 23-27; page 7, line 1-2

We have added new control measurement (Fig. S4) and rewritten the last paragraph as “As shown in Fig. 2e, the ND@pH particles showed pH-dependent release...”.

Page 7, line 3-5

We have added new control measurement (Fig. S5) and rewritten the sentence as “Similarly, the ND@redox particles showed an obvious GSH concentration-dependent release...”.

Page 7, line 19-27; Page 8, line 1-10

We added additional information about the “NV spin population flipped by microwave” in Supplementary Information; added Fig. S6 to illustrate the relative enhancement by comparing the two used pulse scheme; rewritten the last paragraph as “As the spin state has to be manipulated, knowledge about the spin resonance...”.

Page 9, line 1-5

We added a control experiment using ND@pH sensor in Britton-Robinson buffers (Fig. S8), and added the corresponding description as “Using set of Britton-Robinson buffers (with equal composition and ionic strength) we...”.

Page 9, line 8-10

We added additional data sets for Fig. 4d and rewritten the description as “As shown in Fig. 4d, we observed an increase in the measured...”.

Page 9, line 25-27

We added additional statistical view of the ND@pH sensors (Fig. S9), and added the corresponding description as “This is consistent with the statistical view of T_1 observations...”.

Page 10, line 17-28; Page 11, line 1-12

We added new discussion part (as suggested by reviewers) starting from “Although the achieved accuracy (~ 0.7 pH unit) is lower compared to current most sensitive measurement...”.

Page 12, line18-21

We have added the simulation of the NV spin flip probability (Supplementary text and Fig. S10), and corresponding description as “while the linear chirp pulse is much more robust, especially in the presence of inhomogeneous ODMR line broadening...”.

Reviewer #2 (Remarks to the Author):

I appreciate the effort that the authors have put in to address all the concerns that were raised. They have followed up with the requisite control experiments, added supporting data and figures, and added the requested explanations and discussions to the manuscript. I recommend to publish the manuscript.

Dear Dr. Matsiko,

We appreciate the great efforts and valuable time you and the reviewers have taken to comment on our work (original Manuscript ID: NCOMMS-16-17056A).

According to the editorial requests, we have carefully edited the manuscript to comply with the format requirements from Nat. Com..

We would like to publish all the reviewer comments online as a supplementary peer review file. We also approve the draft summary as suggested “The use of nanoscale sensors capable of detection of biological parameters is of great interest in diagnosis. Here, the authors use experimental and theoretical methods to develop a nanodiamond sensor with nitrogen vacancy defects for detection of pH and redox in a microfluidic device.”.

We look forward to your favorable considerations.

Best regards

Zhiqin Chu and Petr Cigler

Reviewers comments:

Reviewer #2 (Remarks to the Author):

I appreciate the effort that the authors have put in to address all the concerns that were raised. They have followed up with the requisite control experiments, added supporting data and figures, and added the requested explanations and discussions to the manuscript. I recommend to publish the manuscript.

Response to reviewer:

We thank reviewer for recommending the current work to publish. There was no specific suggestion on the revision of current work, therefore we did not make any change.